# Construct ceRNA Network and Risk Model of Breast Cancer Using Machine Learning Methods under the Mechanism of Cuproptosis

**DOI:** 10.3390/diagnostics13061203

**Published:** 2023-03-22

**Authors:** Jianzhi Deng, Fei Fu, Fengming Zhang, Yuanyuan Xia, Yuehan Zhou

**Affiliations:** 1Guangxi Key Laboratory of Embedded Technology and Intelligent Information Processing, Guilin University of Technology, Guilin 541006, China; 2College of Information Science and Engineering, Guilin University of Technology, Guilin 541006, China; 3College of Foreign Studies, Guilin University of Technology, Guilin 541004, China; 4College of Pharmacy, Guilin Medical University, Guilin 541199, China

**Keywords:** cuproptosis, breast cancer, PRNP, ceRNA, risk model, machine learning

## Abstract

Breast cancer (BRCA) has an undesirable prognosis and is the second most common cancer among women after lung cancer. A novel mechanism of programmed cell death called cuproptosis is linked to the development and spread of tumor cells. However, the function of cuproptosis in BRCA remains unknown. To this date, no studies have used machine learning methods to screen for characteristic genes to explore the role of cuproptosis-related genes (CRGs) in breast cancer. Therefore, 14 cuproptosis-related characteristic genes (CRCGs) were discovered by the feature selection of 39 differentially expressed CRGs using the three machine learning methods LASSO, SVM-RFE, and random forest. Through the PPI network and immune infiltration analysis, we found that PRNP was the key CRCG. The miRTarBase, TargetScan, and miRDB databases were then used to identify hsa-miR-192-5p and hsa-miR-215-5p as the upstream miRNA of PRNP, and the upstream lncRNA, CARMN, was identified by the StarBase database. Thus, the mRNA PRNP/miRNA hsa-miR-192-5p and hsa-miR-215-5p/lncRNA CARMN ceRNA network was constructed. This ceRNA network, which has not been studied before, is extremely innovative. Furthermore, four cuproptosis-related lncRNAs (CRLs) were screened in TCGA-BRCA by univariate Cox, LASSO, and multivariate Cox regression analysis. The risk model was constructed by using these four CRLs, and the risk score = C9orf163 * (1.8365) + PHC2-AS1 * (−2.2985) + AC087741.1 * (−0.9504) + AL109824.1 * (0.6016). The ROC curve and C-index demonstrated the superior predictive capacity of the risk model, and the ROC curve demonstrated that the AUC of 1-, 3-, and 5-year OS in all samples was 0.721, 0.695, and 0.633, respectively. Finally, 50 prospective sensitive medicines were screened with the pRRophetic R package, among which 17-AAG may be a therapeutic agent for high-risk patients, while the other 49 medicines may be suitable for the treatment of low-risk patients. In conclusion, our study constructs a new ceRNA network and a novel risk model, which offer a theoretical foundation for the treatment of BRCA and will aid in improving the prognosis of BRCA.

## 1. Introduction

Breast cancer, following lung cancer, is the second-most common cancer among women overall [1]. Breast cancer is a heterogeneous illness with unique biological traits, molecular traits, and clinical consequences [2]. Currently, breast cancer is treated with surgery, chemo, radiation, endocrine, and biological targeting therapy. However, the therapeutic impact and prognosis of breast cancer are not optimal, and the recurrence rate and medication resistance of certain patients after therapy are still high. According to the latest statistical research, mortality patterns reflect incidence trends, with BRCA mortality declining more slowly, suggesting that progress in breast cancer research has stalled [3]. The pursuit of biomarkers that aid in the diagnosis, prognosis, and prediction of BRCA has significant implications for monitoring BRCA recurrence and identifying new therapeutic targets during the course of treatment.

As a frequent trace metal, copper plays a significant role in numerous biological processes, including detoxification, iron absorption, mitochondrial respiration, and oxidation resistance. It has been established that Alzheimer’s disease, metabolic syndrome, blood disorders, cardiovascular illnesses, and cancer are all related to the dysregulation of copper homeostasis [4]. Recent research has identified a novel cell death mechanism that is distinct from the conventional cell death pathways, including apoptosis, pyroptosis, and ferroptosis [5]. The pathogenic mechanism is that copper interacts with the fatty acylated parts of the tricarboxylic acid (TCA) cycle directly, which causes excessive fatty acylated protein aggregation and the loss of iron-sulfur cluster proteins, which promotes proteotoxic stress and cell death [5]. Glutathione consumption and the presence of copper ion carriers can promote copper-mediated cell death. Recent research [5] has shown that copper cell death is highly linked to cancer in people, demonstrating that cuproptosis is directly related to the emergence of human cancer. However, the mechanism of BRCA is yet unknown. As a result, we can investigate novel BRCA treatment approaches based on the cuproptosis mechanism to address the drawbacks of conventional therapy.

There is growing evidence according to which non-coding RNAs play critical roles in the incidence, development, and metastasis of many malignancies, thanks to the fast advancement of genomics and transcriptomics. An interaction between RNAs is thought to be mediated by competing endogenous RNA (ceRNA). The miRNAs negatively regulate the expression of their target genes via transcriptional degradation or repression, resulting in gene silencing [6], while ceRNA can regulate gene expression by competitively binding miRNA. The ceRNA disables miRNA by combining miRNA response elements (MREs) with miRNA. The ceRNA regulatory network is crucial in the emergence of various cancers, according to a growing body of research [7], including BRCA [8]. However, the function of ceRNA regulatory networks in BRCA has yet to be well understood. Under the mechanism of cuproptosis, the role of the ceRNA regulatory network in the occurrence and progression of BRCA is almost blank.

Since the concept of cuproptosis, studies on several lncRNAs related to cuproptosis in various human cancers have been continuously carried out. LncRNA is a functional RNA that significantly impacts the development and spread of a variety of malignancies through various processes it mediates [9]. For example, recent studies have confirmed that lncRNA NEAT1 is essential for metabolic changes that promote BRCA growth and metastasis [10]. A study has suggested that lncRNA MALAT1, as a novel oncogenic lncRNA, promotes the progression of BRCA by targeting miR-570-3p [11]. The lncRNA SEMA3B-AS1 targets the miR-3940/KLLN axis to impede the advancement of BRCA [12]. Nonetheless, the role of CRLs in BRCA is still ambiguous. In the recent reports on breast cancer, only two articles [13,14] used cuproptosis-related lncRNAs to construct prognostic models. This suggest that more prognostic models may be needed to explain the role of CRLs in breast cancer.

To sum up, in this study, cuproptosis-related characteristic genes (CRCGs) were screened from the perspective of machine learning. The expression and immune significance of CRCGs in breast cancer were comprehensively clarified. A key CRCG and related ceRNA regulatory networks were identified, and the prospective sensitive medicines of CRGs were screened. Then, we constructed a risk model with prognostic significance based on cuproptosis-related lncRNAs (CRLs). We construct a new ceRNA regulatory network and a novel risk model under the mechanism of cuproptosis, which is the main contribution of this study. It is worth noting that the ceRNA network constructed by us has never been studied before and is very novel. We anticipate that our results will help us further understand the role of CRCGs and CRLs in BRCA and lay a theoretical foundation for BRCA therapy.

## 2. Materials and Methods

### 2.1. The Gathering and Pretreatment of Microarray Data

Data on corresponding BRCA normal and tumor samples were retrieved from Gene Expression Omnibus (GEO) databases (https://www.ncbi.nlm.nih.gov/geo/, accessed on 16 September 2022) and The Cancer Genome Atlas (TCGA) databases (https://portal.gdc.cancer.gov/, accessed on 29 September 2022). The GSE54002, GSE29431, GSE42568, GSE139038, and GSE205185 gene expression profile matrix files were downloaded from the GEO database. The GSE54002, GSE29431, and GSE42568 dataset are all from the GPL570 platform. The GSE54002 dataset contains 433 samples with 16 breast non-tumor samples and 417 breast tumor samples. The GSE29431 contains 66 samples, including 12 breast normal tissue samples and 54 breast tumor tissue samples. The GSE42568 dataset contains 121 samples, with 17 normal breast samples and 104 breast cancer samples. The GSE139038 dataset is from the GPL27630 platform with a total of 65 samples, including 24 normal breast samples (6 of which are apparently normal samples and 18 are paired normal samples) and 41 breast cancer samples. The GSE205185 dataset is from the GPL21185 platform, with a total of 22 samples containing 5 normal breast samples and 17 breast cancer samples. The aforementioned GEO datasets’ series matrix and platform text files were acquired from the GEO database. RNA-Seq data and clinical information from the TCGA-BRCA project from the TCGA database was obtained, and the dataset contains 1226 samples with 113 normal samples and 1113 tumor samples. The platform files’ gene symbols are changed from the matrix files’ gene probe IDs using the Perl (v5.30.0; available at https://strawberryperl.com/, accessed on 29 July 2022) software to obtain a matrix file with the gene name that is recognized internationally. The sva (v3.42.0) R package was then used to merge five datasets from the GEO database, i.e., the gene expression was merged. If we encounter the same gene, we take the mean of the expression of these same genes as the final gene expression. Before further analysis, each dataset was normalized with the limma (v3.50.3) R package. All gene expression data were transformed using the log2 function.

### 2.2. Obtainment, Differential Expression Analysis, and Correlation Analysis of CRGs

A total of 57 CRGs genes were obtained based on prior reports [5,15,16]. Next, the limma package was used for differential expression analysis of the combined GEO expression data, and the same operation was performed for TCGA-BRCA. It is worth noting that the data in TCGA-BRCA are normalized to fragments per kilobase of transcript per million (FPKM). The screening criteria for differential expression genes (DEGs) was set as |logFC| > 2 and *p* < 0.05. Finally, the corrplot (v0.92) R package was used to visualize the correlation of 39 DEGs.

### 2.3. Screening of Cuproptosis-Related Characteristic Genes

According to previous studies [17], the least absolute shrinkage and selection operator (LASSO), recursive feature elimination by support vector machine (SVM-RFE [18]), and random forest (RF) methods can be used to screen characteristic genes. Here, we hypothesized that these three machine learning methods could also screen out characteristic genes from cuproptosis-related genes and obtain meaningful biomarkers in subsequent analysis. Therefore, these three machine learning methods were applied for the feature selection of CRGs to screen out cuproptosis-related characteristic genes (CRCGs). Compared with regression analysis, LASSO is a dimension reduction method that is superior in evaluating high-dimensional data. The LASSO analysis was implemented by constructing a penalized function with 10-fold cross-validation via the glmnet (v4.1-4) R package. SVM-RFE is superior to linear discriminant analysis (LDA) and the mean squared error (MSE) in selecting relevant features and eliminating redundant features, for this was applied for feature selection of CRGs via the e1071 (v1.7-11) package with 10-fold cross-validation. The RF algorithm, an approach of supervised machine learning, was used to rank the CRGs via randomForest (v4.7-1.1). We set the random forest tree to 500, and the predictive performance was estimated via 10-fold cross-validation. Eventually, the differential visualization of 14 CRCGs in the form of heatmaps was performed using pheatmap (v1.0.12) and ggpubr (v0.4.0) R packages, besides using the RCircos (v1.2.2) R package to show the location of CRCGs on chromosomes.

### 2.4. GO, KEGG Enrichment Analysis

The clusterprofiler (v4.2.2) R package was primarily used to conduct the GO and KEGG enrichment analysis. The GO database was utilized to examine these CRCGs’ biological characteristics, and the signaling pathway of CRCGs was detected via the KEGG database. *p* < 0.05 was used as the screening condition to gain the main GO enrichment function and KEGG pathway.

### 2.5. Immune Cell Infiltration Analysis and Correlation Analysis with CRCGs

We utilized the CIBERSORT algorithm in the CIBERSORT script to determine the fraction of immune infiltrating cells in each sample. Correlation analyses between the CRCGs and immune cells were conducted by the Pearson method. Finally, immune cell differences were plotted using the vioplot (v0.3.7) R package, and correlations between CRCGs and immune cells were visualized using the ggplot2 (v3.3.6) R package.

### 2.6. Identification of Key CRCG and Construction of ceRNA Regulatory Network

The protein-protein interaction (PPI) network of 14 CRCGs was constructed using the STRING database (https://cn.string-db.org/, accessed on 10 October 2022) and Cytoscape software (v3.7.2; https://cytoscape.org/, accessed on 10 October 2022), and the MCC algorithm of the cytoHubba plugin in Cytoscape software screened the hub CRCGs. After the hub CRCGs were identified, a key CRCG was identified by combining the correlation between CRCGs and immune cells. TargetScan (https://www.targetscan.org/, accessed on 10 October 2022), miRTarBase (https://mirtarbase.cuhk.edu.cn/, accessed on 10 October 2022), and miRDB (http://www.mirdb.org/, accessed on 10 October 2022) databases were used to predict miRNA targets associated with the key CRCG. Then, the intersection of the prediction results of these three databases was considered as the key miRNA targets associated with the key CRCG. Next, lncRNA targets linked to the key miRNAs were predicted using the StarBase database. Finally, we constructed a biologically significant regulatory network of ceRNA based on the prediction results obtained above, combined with the expression level in TCGA-BRCA.

### 2.7. Identification of CRLs and Construction of the Risk Model

Firstly, mRNA and lncRNA in TCGA-BRCA data were extracted, and then the limma package was used to calculate the correlation between 57 CRGs and lncRNAs. In total, 85 CRLs were obtained with correlation coefficients |R| > 0.3 and *p* < 0.05 as screening criteria. Next, combined with the expression profile of CRLs and the survival time of patients, we obtained prognosis-related CRLs for BRCA patients using univariate Cox regression analysis and evaluated their prognostic value. The CRLs with prognostic value were randomly split into training and test sets at a ratio of 1:1, and predictors were chosen using LASSO regression analysis to prevent overfitting. Finally, a risk model of the CRLs with prognostic value was built in the training set using multivariate Cox regression analysis. The risk score of CRLs with prognostic value was calculated using the following formula:(1)$$RiskScore=\sum_{i=1}^{N}\left(E_{i}*C_{i}\right)$$

In the formula, the variables *N*, *E*${}_{i}$, and *C*${}_{i}$ stand for the number of CRLs having a prognostic value in the risk model, the expression value of each CRL, and the regression coefficient of each CRL in multivariate Cox regression analysis, respectively. Patients were divided into high-risk group and low-risk group according to the median risk score of the calculated results. The viability of the high- and low-risk groups was then assessed using Kaplan–Meier curves. Independent prognostic variables were then screened out using univariate Cox regression analysis and multivariate Cox regression analysis, and the model’s accuracy was assessed using the receiver operating characteristic curve (ROC) curve and concordance index (C-index).

### 2.8. Screening of Potentially Sensitive Medicines and Analyzing the Correlation between Drug Sensitivity and Risk Score

To better apply this model to clinical treatment, we used the pRRophetic (v0.5) R package to calculate the half maximal inhibitory concentration (IC50) of anti-BRCA medicines. Then, two built-in data sets from the pRRophetic package, cgp2016ExprRma and PANCANCER_IC_Tue_Aug_9_15_28_57_2016, were used to screen out potentially sensitive medicines in BRCA patients, and the screening condition was *p* < 0.001. Next, the correlation between risk scores and sensitivity of potential medicines was visualized by ggplot2 and ggpubr packages. In addition, differential expressions of drug sensitivity in the high- and low-risk groups were presented as shown in boxplots.

### 2.9. Statistical Analysis

The correlation of CRGs in BRCA samples was evaluated using Pearson’s correlation test. To determine the level of significance between the two groups, Wilcoxon tests or Student *t*-tests were used, using the log-rank test to assess the significance between Kaplan-Meier survival curves. To determine the hazard ratios (HRs) and 95% confidence intervals (CIs) for the risk scores and other key clinical indicators, we also conducted univariate and multivariate Cox regression analysis. R (v 4.1.2; https://www.r-project.org/, accessed on 16 July 2022) software was used to conduct all statistical analyses. ***, *p* < 0.001; **, *p* < 0.01; *, *p* < 0.05. *p* < 0.05 was considered statistically significant.

## 3. Results

The flow chart of the whole study is displayed in (Figure 1). It is worth noting that two databases, GEO and TCGA, were used in this study. The five datasets in the GEO database are mainly used for screening key CRCG and constructing related ceRNA regulatory networks. At the same time, TCGA-BRCA was used to verify the expression of PRNP and lncRNA CARMN in ceRNA in breast cancer, as well as to construct a risk model by CRLs with prognostic value.

### 3.1. Differential Expression Analysis and Correlation Analysis of CRGs

Firstly, we divided the data into an experimental (tumor) group and a control (normal) group through the sample information from the five GEO datasets. The results of differential analysis (Figure 2A) showed that 39 out of 57 CRGs were differentially expressed in tumor samples compared to normal samples. Genes such as ARF1, SLC25A5, and CDKN2A were significantly up-regulated in tumor samples, while genes such as SNCA, PRNP, and GLS were significantly down-regulated in tumor samples. Correlation analysis (Figure 2B) showed that MT1H and MT2A had the most significant positive correlation, while CDKN2A and SLC22A5 had the most significant negative correlation.

### 3.2. Selection of CRCGs via LASSO, SVM-RFE, and RF Methods

For screening out characteristic genes among 39 CRGs, we used three methods, i.e., LASSO, SVM-RFE, and RF. The 10-fold cross-validation result shows that the optimal lambda for the LASSO method was 0.003 (Figure 3A,B). Next, we constructed a LASSO model with the highest accuracy. This resulted in 28 characteristic genes being found, including SLC25A5, NDUFB2, PDHX, DLST, CCS, SLC22A5, CCDC22, PDHA1, FDX1, MT2A, MT1E, MT1F, MT1G, MT1H, MT1X, LIAS, LIPT1, PRNP, GLS, SNCA, PDHB, BACE1, SLC31A2, CDKN2A, NFE2L2, ATP7A, and BECN1. For the SVM-RFE method, when the feature number was 27, the accuracy (0.975) of the classifier was the highest, which means the error was the smallest at this time (Figure 3C,D). Therefore, 27 feature genes are screened, containing MT1X, MT2A, CDKN2A, MT1E, SLC25A5, DLST, MT1F, NFE2L2, PRNP, BECN1, CCDC22, PDHB, GLS, BACE1, FDX1, CCS, ATP7A, MT1G, ATOX1, LIAS, SNCA, SLC22A5, PRND, NDUFB2, DLAT, NDUFA1, ATP7B, and NDUFB1. Furthermore, the third method, the RF algorithm, identified 24 characteristic genes with relative importance greater than 2 as the screening condition (Figure 3E,F): SNCA, CDKN2A, FDX1, BACE1, MT1X, PRNP, SLC6A3, ARF1, DAXX, NFE2L2, SLC25A5, LIPT1, LIAS, SLC31A2, BECN1, CYP1A1, MT1E, DLST, PDHX, DLD, PDHA1, PDHB, GLS, and ATOX1. A total of 14 CRCGs were produced by combining the screening findings from the aforementioned three methods (Figure 3G), namely SLC25A5, DLST, FDX1, MT1E, MT1X, LIAS, PRNP, GLS, SNCA, PDHB, BACE1, CDKN2A, NFE2L2, and BECN1. Therefore, these three machine learning methods can screen out CRCGs, which proves the validity of our previous hypothesis. Finally, we show the position of 14 CRCGs on chromosomes by a loop graph (Figure 3H).

### 3.3. Differential Expression Analysis, GO, and KEGG Enrichment Analysis of CRCGs

We first analyzed the differential expression of 14 CRCGs in five GEO datasets, and the results (Figure 4A,B) displayed that SLC25A5 and CDKN2A were significantly up-regulated in tumor samples. DLST, FDX1, MT1E, MT1X, LIAS, PRNP, GLS, SNCA, PDHB, BACE1, NFE2L2, and BECN1 were significantly down-regulated in tumor samples. We subsequently analyzed the GO functional enrichment and KEGG pathway of these 14 CRCGs to clarify the potential roles of these CRCGs. The findings demonstrated that the 14 CRCGs were mostly enriched in the cellular response to copper ion (GO:0071280), response to copper ion (GO:0046688), mitochondrial matrix (GO:0005759), terminal bouton (GO:0043195), copper ion binding (GO:0005507), and iron-sulfur cluster binding (GO:0051536) via GO functional enrichment analysis (Figure 4C–E). Additionally, the KEGG pathway analysis (Figure 4F,G) indicated that the 14 CRCGs mainly participated in Alzheimer’s disease (hsa05010), the pathways of multiple neurodegeneration diseases (hsa05022), Parkinson’s disease (hsa05012), citrate cycle (TCA cycle) (hsa00020), mineral absorption (hsa04978), central carbon metabolism in cancer (hsa05230), and carbon metabolism (hsa01200).

### 3.4. Identification of Key CRCG and Immune Cell Infiltration Analysis

We constructed the PPI network of 14 CRCGs through the STRING database and then sorted the 14 CRCGs through the MCC algorithm of cytoHubba plug-in Cytoscape software. The results (Figure 5A) show that SNCA, PRNP, and BECN1 were hub genes. These three hub genes can be used as breast cancer diagnostic genes, and their diagnostic performance was evaluated using AUC (Appendix A). The results of immune infiltration (Figure 5B) reveal that immune cells such as macrophages M0 and M1 were more infiltrated in the tumor group compared to the normal group, whereas immune cells such as macrophages M2 and resting mast cells were less infiltrated in the tumor group. Next, we conducted the correlation analysis on 14 CRCGs and immune cells, and the findings (Figure 5C) reveal that PRNP was significantly correlated with 13 different types of immune cells. BECN1 and CDKN2A were next, with significant correlations to 11 and 10 different types of immune cells, respectively. Based on the above analyses, we identified PRNP as the key CRCG.

### 3.5. Construct and Verify PRNP-Related ceRNA Regulatory Network

PRNP was the key CRCG identified according to the above. We then used three online databases, TargetScan, miRDB, and miRTarBase, to screen PRNP-related miRNAs in BRCA. In total, 569 PRNP-related miRNAs were screened from the TargetScan database. 109 PRNP-related miRNAs were screened from the miRDB database. Furthermore, 57 PRNP-related miRNAs were screened from the miRTarBase database. By taking the intersection (Figure 6A) of the screening results of the three databases, 10 PRNP-related miRNAs, namely hsa-miR-148a-3p, hsa-miR-148b-3p, hsa-miR-221-5p, hsa-miR-8073, hsa-miR-495-3p, hsa-miR-5688, hsa-miR-215-5p, hsa-miR-192-5p, hsa-miR-188-3p, and hsa-miR-3156-3p, were obtained. Before screening these 10 miRNA-related lncRNAs, we verified the expression of PRNP and these 10 miRNAs in BRCA. The expression of PRNP in the tumor group of TCGA-BRCA was significantly underexpressed (Figure 6C), which was consistent with the expression of PRNP in the GEO database. Through the online database dbDEMC (https://www.biosino.org/dbDEMC/, accessed on 12 October 2022), we verified the expression of these 10 miRNAs in breast cancer and obtained two miRNAs, i.e., hsa-miR-192-5p and hsa-miR-215-5p that were overexpressed in breast cancer (Figure 6D,E). Obviously, these two miRNAs have a negative regulatory relationship with PRNP. Then, we used the StarBase database to screen lncRNAs related to the hsa-miR-192-5p and hsa-miR-215-5p in breast cancer and found that 31 lncRNAs had a regulatory relationship with these two miRNAs. The miRNA-lncRNA regulatory network diagram (Figure 6G) was also drawn. According to the scientific hypothesis of the ceRNA regulatory network, we next looked for lncRNAs that were underexpressed in breast cancer. Therefore, down-regulated lncRNA in tumor samples were screened out from lncRNAs with differential expression of TCGA-BRCA, and then these lncRNAs will be intersected with those 31 lncRNAs to obtain one lncRNA (Figure 6B), which was CARMN. Next, we verified CRAMN’s expression in the TCGA-BRCA dataset, and the results (Figure 6F) showed that CARMN was significantly underexpressed in tumor samples. Finally, we constructed a new ceRNA regulatory network (Figure 6H) associated with BRCA, namely mRNA PRNP/miRNA hsa-miR-215-5p and hsa-miR-192-5p/lncRNA CARMN. This further confirms our previous hypothesis.

### 3.6. Identified CRLs and Constructed Risk Model

First, we conducted a co-expression analysis of CRGs and lncRNAs in TCGA-BRCA, the visualized results of which are presented by Sankey diagrams in the Appendix A. The correlation coefficient of |R| > 0.3 and *p* < 0.05 was set as the screening threshold, and 85 CRLs were finally identified. Then, the expression profiles of these 85 CRLs were combined with clinical information. All samples were split into training and test sets at a ratio of 1:1, and a univariate Cox regression analysis was carried out in the training sets to assess the prognostic significance of these CRLs (Figure 7A). Then, we used LASSO (Figure 7B,C) to reduce the feature dimension of these CRLs, in which lambda was equal to 0.021, and the effect was optimal. A total of 7 CRLs with significant prognostic value were obtained, namely C9orf163 (*p*-value = 0.045, HR = 2.556), THBS3-AS1 (*p*-value = 0.038, HR = 0.306), PHC2-AS1 (*p*-value = 0.004, HR = 0.088), ZNF197-AS1 (*p*-value = 0.039, HR = 0.225), AC087741.1 (*p*-value = 0.035, HR = 0.534), AC073569.3 (*p*-value = 0.049, HR = 0.115), and AL109824.1 (*p*-value = 0.008, HR = 1.785). Then, a risk model linked to CRLs with prognostic value was built using multivariate Cox regression analysis, and four CRLs were screened, namely C9orf163, PHC2-AS1, AC087741.1, and AL109824.1; (Figure 7D) shows the correlation between CRGs and these four CRLs with prognostic value. The four CRLs and their corresponding regression coefficients are shown in Appendix A. According to the formula we set up before, we can derive the risk score = C9orf163 * (1.8365) + PHC2-AS1 * (−2.2985) + AC087741.1 * (−0.9504) + AL109824.1 * (0.6016). We used the risk score formula to calculate the risk score of all breast cancer patients and the median risk score. According to the median risk score, all breast cancer patients were split into high- and low-risk groups. After performing a survival analysis (Figure 8A–C) on samples from the high- and low-risk groups, overall survival (OS) curves were drawn, and deletions at different time points during follow-up were shown. The findings demonstrated that high-risk patients had considerably poorer prognoses than low-risk patients, with the median survival times of high-risk patients in the training and test groups falling short of 10 years. Patients had a greater chance of dying and a shorter survival time as their risk scores increased. Figure 8D–I shows the length of survival and current health of breast cancer patients with increased risk scores. Finally, the constructed risk model is then put to the test using the test sets. Patients in the low-risk group in the test sets have lower risk scores and live longer than those in the high-risk group, as shown by the survival curves. Using the previously mentioned method again, risk curves and scatter plots were utilized to display survival time and survival status. The findings demonstrate that as the risk score rose, so did the patient’s mortality risk. The test set’s findings and the training set’s results agree, proving that the risk model constructed by these four CRLs is reliable.

### 3.7. Independent Prognostic Analysis and Building a Nomogram

To determine whether the risk model could be employed as an independent prognostic factor for breast cancer, univariate and multivariate Cox regression analyses were carried out. Three independent prognostic factors, namely age, stage, and risk score, were discovered by univariate Cox regression analysis (Figure 8J) and were all shown to be significantly linked with OS. Fortunately, the results of multivariate Cox regression analysis (Figure 8K) obtained consistent results with the univariate Cox regression analysis. Age, stage, and risk score were all significantly correlated with OS. This demonstrates that the risk model we constructed can be seen as an independent prognostic factor. Following that, ROC curves and area under the curve (AUC) were used to assess the specificity and sensitivity of the risk model for the prognosis of BRCA. The findings reveal that the AUC of 1-, 3-, and 5-year OS was, respectively, 0.721, 0.695, and 0.633 in all samples (Figure 9A). The AUC of the 1-, 3-, and 5-year OS was, respectively, 0.740, 0.776, and 0.715 in the training set samples (Figure 9B). The AUC of the 1-, 3-, and 5-year OS was, respectively, 0.697, 0.604, and 0.550 in the test set samples (Appendix A). In addition, both the AUC values and the risk model’s C-index were higher than the clinical traits (Figure 9C,D). Finally, to better forecast the 1-, 3-, and 5-year survival rates of breast cancer patients, we combined the clinical features and the risk model to create a nomogram (Figure 9E). The accuracy of the nomogram was confirmed using the calibration curve (Figure 9F). The outcomes demonstrate that the actual observed value and the predicted value may be well matched, demonstrating the excellent accuracy of the nomogram.

### 3.8. Drug Sensitivity Analysis

The cgp2016ExprRma and PANCANCER_IC_Tue_Aug_9_15_28_57_2016, two built-in datasets in the pRRophetic software package, were used to identify potential medicines in connection with the onset and progression of BRCA. Following the screening, we discovered 50 drugs that had a significant correlation to risk score, with 17-AAG (Figure 10A) having a negative correlation with the risk score. Forty-nine medicines, including, among others, 5-Fluorouracil, AP-24534, BAY 61-3606, Cytarabine, Epothilone B, Bleomycin (50 uM), and BI-2536 (Figure 10B–H), showed a positive correlation with the risk score. We then verified the differential expression of these 50 potential agents in the high- and low-risk groups. The results reveal that, in addition to the significantly underexpressed 17-AAG (Figure 10I) in the high-risk group, forty-nine other medicines, including 5-Fluorouracil, AP-24534, BAY 61-3606, Cytarabine, Epothilone B, Bleomycin (50 uM), and BI-2536 (Figure 10J–P), were all significantly overexpressed in the high-risk group. These findings imply that 17-AAG may be used as a therapeutic medicine for people at high risk. In contrast, the other 49 medicines may be suitable for treating low-risk patients. The correlation between 50 medicines and risk score, as well as the differential expression of these medicines in the high- and low-risk group, is shown in the Appendix A.

## 4. Discussion

Breast cancer currently has a dismal prognosis and treatment, making it the second most frequent kind of cancer that threatens women. Therefore, a thorough research of the pathogenesis of BRCA is necessary to improve its prognosis and promote precision medicine. Cuproptosis-related genes are crucial in regulating breast cancer progression, prognosis, immune cell infiltration, and response to immunotherapy [19]. Therefore, we explored the regulatory role of a key CRCG in constructing a ceRNA network under the mechanism of cuproptosis, and in order to improve the prognosis of breast cancer and promote precision medicine, a risk model based on four CRLs was constructed.

First, we conducted a differential expression study on 57 CRGs, which revealed that 39 CRGs’ mRNA levels had a significantly differential expression in tumor samples and normal samples. Through the screening of three machine learning methods, LASSO, SVM-RFE, and RF, 14 characteristic genes were obtained, namely SLC25A5, DLST, FDX1, MT1E, MT1X, LIAS, PRNP, GLS, SNCA, PDHB, BACE1, CDKN2A, NFE2L2, and BECN1. SLC25A5 and CDKN2A were found to be significantly up-regulated in tumor samples after we conducted differential analysis on these 14 CRCGs. However, DLST, FDX1, MT1E, MT1X, LIAS, PRNP, GLS, SNCA, PDHB, BACE1, NFE2L2, and BECN1 were significantly down-regulated in tumor samples. Some studies have shown that SLC25A5 is a gene related to lipid metabolism, which is closely connected to BRCA patients’ prognoses [20]. CDKN2A is crucial in the immunotherapy of triple negative breast cancer (TNBC) and can be acted as a prognostic factor of TNBC [21]. In TNBC, the depletion of DLST inhibits cell growth and induces cell death [22]. The expression of LIAS is related to hypoxia, angiogenesis, and DNA repair, and the high expression of LIAS in lung cancer patients is adverse to the prognosis of patients [23]. SNCA enhances sensitivity to commonly used anti-tumor agents and immune cell infiltrating and prevents EMT and metastasis of BRCA [24]. NFE2L2 was abnormally expressed in human pan-cancer and highly linked with the degree of DNA methyltransferase expression and mismatch repair (MMR) gene mutation [25]. The decrease in BECN1 degradation induced by SLC9A3R1 resulted in enhanced autophagy stimulating activity of breast cancer cells [26]. These findings suggest that the 14 CRCGs were somewhat involved in the emergence of breast cancer and other cancer types. These 14 CRCGs were mainly enriched in GO functions, such as cell response to copper ions, indicating that these CRCGs were closely related to the process of cell death induced by copper ions. The enrichment of CRCGs in some disease pathways, such as Alzheimer’s pathways, suggests that these 14 CRCGS are associated with Alzheimer’s disease and other diseases. In in vitro and mouse xenografts, the accumulation of exogenous or endogenous citrate in the cytoplasm dramatically increased the motility, invasion, and metastasis of hypoxic TNBC cells [27].

According to the results of immune infiltration, we found that, compared with the normal group, there was more infiltration of macrophages M0, macrophages M1, and other immune cells within the tumor group. In comparison, there was less infiltration of macrophages M2 and mast cells resting. In breast cancer, DRD2 regulates the microenvironment because it promotes the M1-polarization of macrophages and triggers thermal apoptosis performed by GSDME [28]. Macrophage M2 polarization was inhibited by PM37, and radiation resistance of IBC (inflammatory breast cancer) was prevented by down-regulating PRKCZ [29]. Macrophage M2 produces CHI3L1, which promotes the spread of breast and stomach cancer cells both in vitro and in vivo [30]. In comparison to the high-risk group for breast cancer, the proportion of stationary resting dendritic and mast cells was higher in the low-risk group [31], which is consistent with our immune infiltration analysis results.

The arrangement and mapping of PPI networks allow for further knowledge to be extracted about the evolutionary relationships between species [32]. By constructing the PPI network for 14 CRCGs, we found that the hub CRCGs are SNCA, PRNP, and BECN1. Combined with the correlation of CRCGs with immune infiltration, we identified a key CRCG, PRNP. In neuroblastoma cells, IR activates ATM, causes JNK to be phosphorylated by TAK1 in a manner that then activates AP-1 transcription factor, which in turn increases the transcriptional activity of the PRNP promoter by interacting with AP-1 binding sites [33]. Experimental investigation [34] has presented that the expression of PrPC, including p53 ag and the prion protein coding gene (PRNP), are intimately linked to the development and spread of malignancies (tumor suppressor gene). It is interesting to note that this matches the outcomes of our TCGA-BRCA validation, with a low expression of PRNP in tumor samples. Next, we constructed PRNP-related ceRNA regulatory networks, namely mRNA PRNP/miRNA hsa-miR-192-5p and hsa-miR-215-5p/lncRNA CARMN. It was verified by the dbDEMC database that these two miRNAs were significantly up-regulated in BRCA tumor samples, while CARMN was significantly down-regulated in BRCA tumor samples. Therefore, this regulatory network satisfies the ceRNA scientific hypothesis. A recent study [35] has found that miR-192-5p is significantly up-regulated in LABC (locally advanced breast cancer) patients, which is consistent with our verification results. However, hsa-miR-215-5p’s expression in BRCA was not found in previous reports. Encouragingly, studies [36] have shown that CARMN is down-regulated at different stages of malignant transformation of breast tissue and can also inhibit the occurrence of TNBC tumors and improve the sensitivity to cisplatin. PRNP is an ER stress regulatory gene, which can raise BRCA’s survival rate [37]. No somatic mutation of the PRNP gene has been found in breast cancer and glioblastoma, indicating that PRNP is a tumor suppressor gene [38]. According to the PRNP transcript levels in tumor specimens taken before treatment classified as relapse-free survival following neoadjuvant anthracycline therapy, it is known from this study that PRNP is associated with breast cancer [39]. Therefore, in summary, the mRNA PRNP/miRNA hsa-miR-192-5p and hsa-miR-215-5p/lncRNA CARMN regulatory networks discovered in our study are very promising to offer a theoretical foundation for inhibiting the occurrence and development of BRCA. Moreover, it is worth noting that this regulatory network has not been reported before.

Four CRLs with prognostic value were found by univariate Cox, LASSO, and multivariate Cox regression analysis, namely C9orf163, PHC2-AS1, AC087741.1, and AL109824.1. The construction of a novel risk model employing these four CRLs was followed by the verification that the risk model may function as an independent prognostic factor. We then used the ROC curve to assess the specificity and sensitivity of this model for predicting the prognosis of breast cancer. The findings reveal that the 1-, 3-, and 5-year AUC of all samples was, respectively, 0.721, 0.695, and 0.633. The performance of the risk model was then compared to that of other clinical traits using the ROC curve and C-index, and the findings indicate that the risk model performed better than the majority of clinical traits. Finally, we construct a nomogram with high accuracy to help the diagnosis of breast cancer. Cuproptosis-related lncRNAs’ prognostic model has only been reported in a few studies [13,14]. In these two studies, 11 and 37 CRLs were used to construct prognostic models of breast cancer, while only four CRLs were used in this study. It is worth noting that fewer CRLs were used as predictors in this study, but the accuracy of prediction was not much different from that in studies [13,14]. The comparison of the three models are shown in Appendix A. It can be seen that the risk model we built is not only simple, but also has good prediction performance. This shows that the risk model is more in line with people’s expectations and has greater application value. Therefore, the risk model we constructed is expected to provide theoretical support for the treatment of breast cancer. In addition, this risk model may be applied to clinical practice. For example, some studies [40,41] developed clinical calculators. These calculators may be used to forecast patient outcomes and could be helpful to doctors when formulating treatment plans. Therefore, if validated by experiments, the risk model constructed in this study may be used clinically to assess the risk of breast cancer patients and predict the prognosis of patients.

Finally, we screened 50 medicines that had a significant correlation with a risk score and confirmed how these medicines differed in their expression between the high- and low-risk groups. Except for 17-AAG, which was considerably underexpressed in the high-risk group, forty-nine medicines, including 5-Fluorouracil, AP-24534, BAY 61-3606, Cytarabine, Epothilone B, BI-2536, and Bleomycin (50 uM), were all significantly overexpressed in the high-risk group. Therefore, we conjecture that 17-AAG may be a therapeutic agent for high-risk patients. In contrast, the other 49 medicines may be suitable for treating low-risk patients. 17-AAG, a competitive inhibitor of heat shock protein Hsp90, has an inhibitory effect on tumor development. SAL, along with 17-AAG, induces apoptosis, inhibits autophagy, and has a synergistic inhibitory impact on the development of breast cancer cells [42]. Treatment with minimally hazardous doses of 17-AAG and SQD triggered apoptosis and increased the suppression of BC cell proliferation [43]. Interestingly, this is consistent with our findings that 17-AAG could be an effective treatment for breast cancer patients. TQ can help 5-Fluorouracil work more effectively against TNBC cells by coordinating its anticancer effects [44]. Mcl-1 expression in breast cancer cells is down-regulated by BAY 61-3606, making cancer cells more susceptible to TRAIL-mediated apoptosis [45]. It is feasible for LM patients with breast cancer to respond rapidly to the quick start of intracellular liposomal cytarabine treatment [46]. BI-2536 is a mitotic inhibitor that blocks the growth and invasion of cancer cells [47]. BI-2536 was demonstrated to decrease tumor development and metastasis in vivo using TAMR-MCF-7 cells injected into xenografts and spleen-liver metastasis models [48]. This shows that most of the medicines we screened have a significant role in the treatment of BRCA, and these 50 medicines provide more options for treating BRCA.

In summary, we explored the role of CRGs and CRLs in breast cancer. Through a series of bioinformatics analyses, this study identified PRNP as a key CRCG, identified the function and pathways of CRCGs enrichment, and obtained four prognostic CRLs and 50 drugs related to breast cancer treatment. Most importantly, this study constructs a new ceRNA network and a novel risk model, which has not been seen in previous studies. However, our study also has some shortcomings. The ceRNA regulatory network we constructed has not been verified by biological experiments, and whether the risk model has practical application value needs to be verified by relevant clinical trials. Therefore, future work should include two aspects: On the one hand, to verify the application value of the ceRNA network and risk model constructed in this study in biomedicine. On the other hand, improving the machine learning methods used in this study could lead to the development of more novel biomarkers and prognostic models.

## 5. Conclusions

In conclusion, we combined machine learning and bioinformatics methods to conduct the analysis in this study. A total of 14 CRCGs were screened using three machine learning methods, and then a key CRCG was identified. More importantly, we developed a new ceRNA regulatory network related to this key CRCG. In addition, we constructed a novel risk model through four CRLs. Therefore, the key contributions of this work are that we developed a new ceRNA regulatory network and a novel risk model. It is important to note that the ceRNA network that we constructed is rather new and has never been investigated previously. This study provides strong theoretical support for the discovery of new treatment strategies that may improve the prognosis of BRCA and decrease the mortality of patients.

## Figures and Tables

**Figure 1 diagnostics-13-01203-f001:**
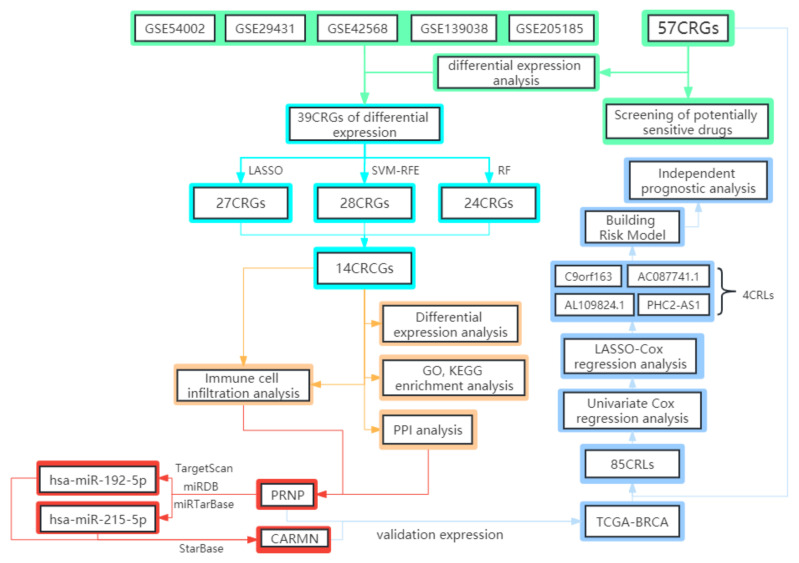
The flow chart of this study.

**Figure 2 diagnostics-13-01203-f002:**
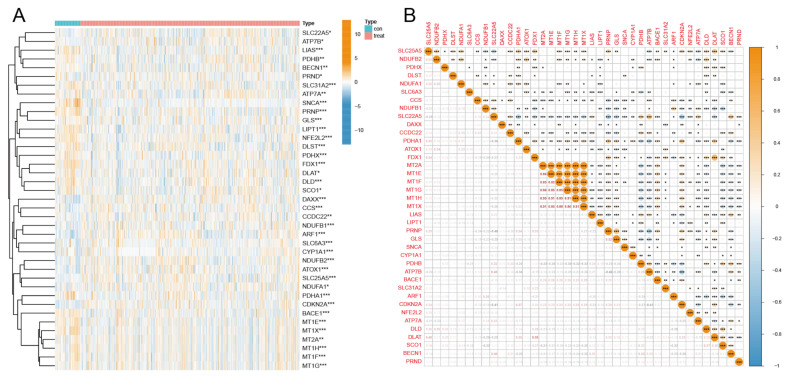
(**A**) 39 CRGs differentially expressed in breast cancer tumor samples compared to normal samples in GEO. Orange means up-regulated, and chrysanthemum blue means down-regulated. The darker the color, the more significant the difference. (**B**) Correlation between 39 cuproptosis-related genes. Orange (positive) indicates a positive correlation between genes, while chrysanthemum blue (negative) indicates a negative correlation. ***, *p* < 0.001; **, *p* < 0.01; *, *p* < 0.05. *p* < 0.05 was considered statistically significant.

**Figure 3 diagnostics-13-01203-f003:**
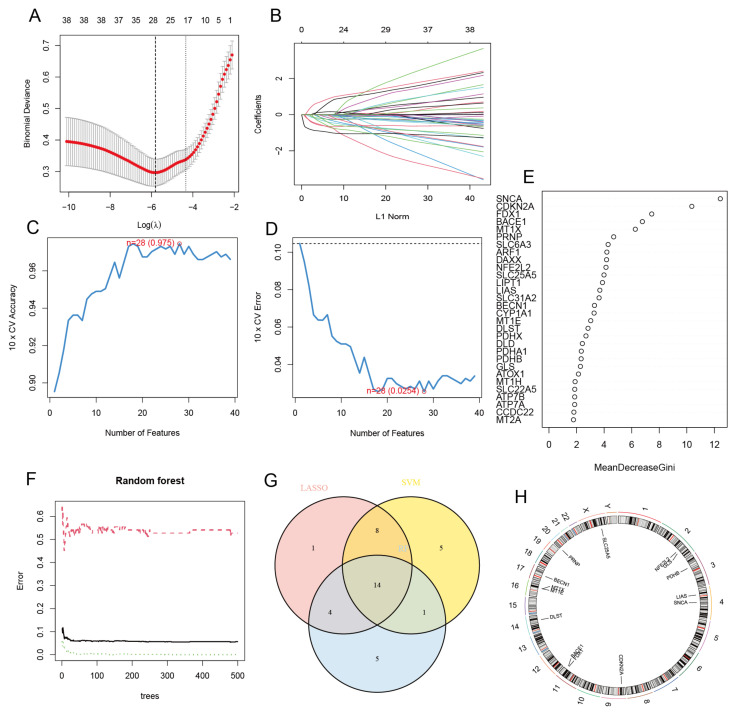
Selection of cuproptosis-related characteristic genes and visualization of the position of 14 CRCGs on chromosomes: (**A**) LASSO coefficient profiling. A solid hammer line represents binomial deviance. A bold dotted line indicates the optimal lambda value. (**B**) Each curve, i.e., each color represented a feature corresponding to a gene. (**C**) 10-fold cross-validation was used for feature dimension reduction. When *n* = 28, the accuracy is the highest, i.e., (**D**) the error is the smallest. (**E**) The CRCGs are ranked according to their relative importance. The first 30 CRCGs are shown here. (**F**) The relationships between the quantity of trees and the error rate in random forest. (**G**) Venn diagram. The CRCGs were screened out via LASSO, SVM-RFE, and RF algorithms. (**H**) The position of 14 CRCGs on chromosomes.

**Figure 4 diagnostics-13-01203-f004:**
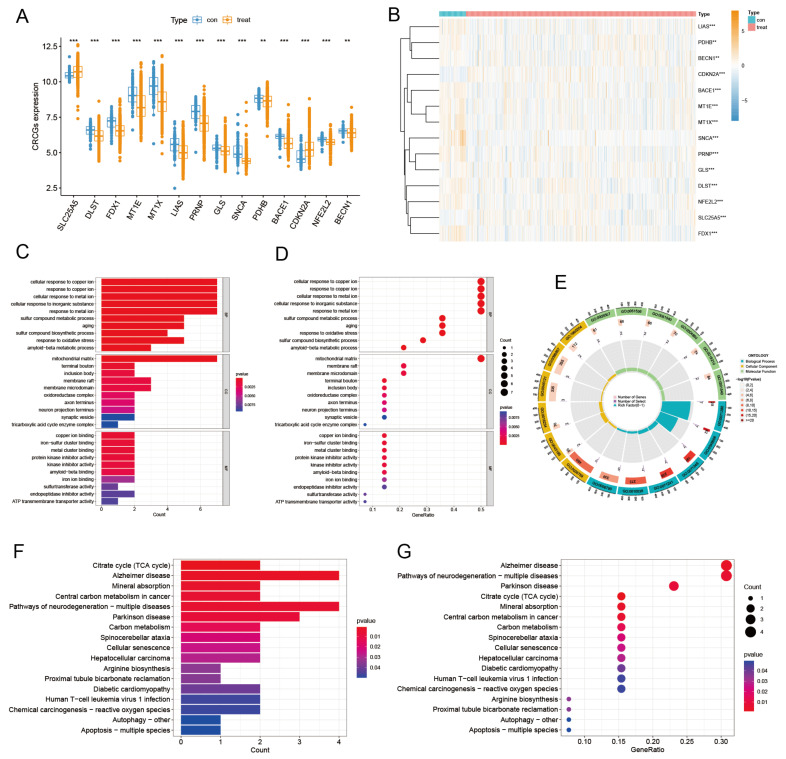
Differential expression analysis of CRCGs in GEO and enrichment analysis. (**A**) Boxplot of differential expression of 14 CRCGs in tumor group and normal group, and (**B**) heat map. (**C**) Barplot (**D**), Bubble plot, and (**E**) loop graph of GO enrichment analysis results of 14 CRCGs in breast cancer. (**F**) Barplot (**G**) and Bubble plot of KEGG enrichment analysis results of 14 CRCGs in breast cancer. ***, *p* < 0.001; **, *p* < 0.01. *p* < 0.05 was considered statistically significant.

**Figure 5 diagnostics-13-01203-f005:**
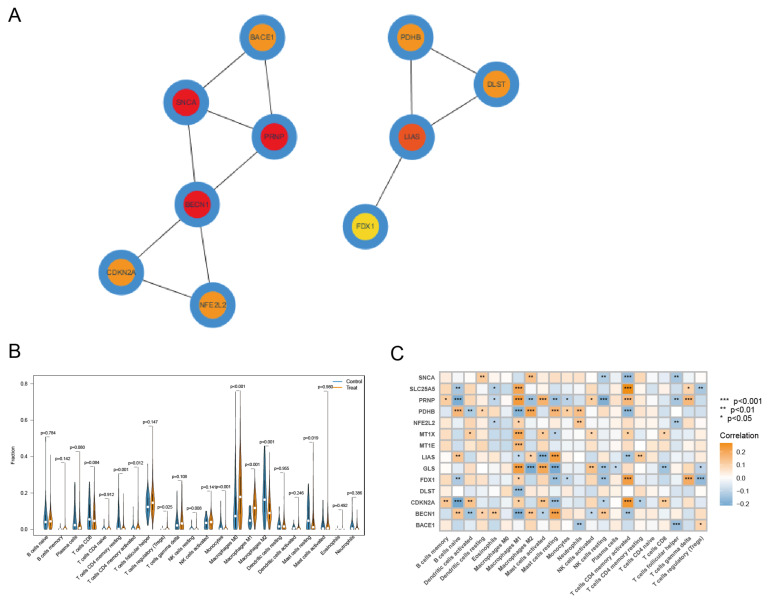
PPI network, immune infiltration analysis of CRGs. (**A**) Interaction network of the first ten CRCGs. The darker the color, the more important the gene. Red represents the hub gene. (**B**) The relative content of immune cells in tumor and normal group. (**C**) Correlation between 14 CRCGs and immune cells, and the darker the color, the more significant the correlation.

**Figure 6 diagnostics-13-01203-f006:**
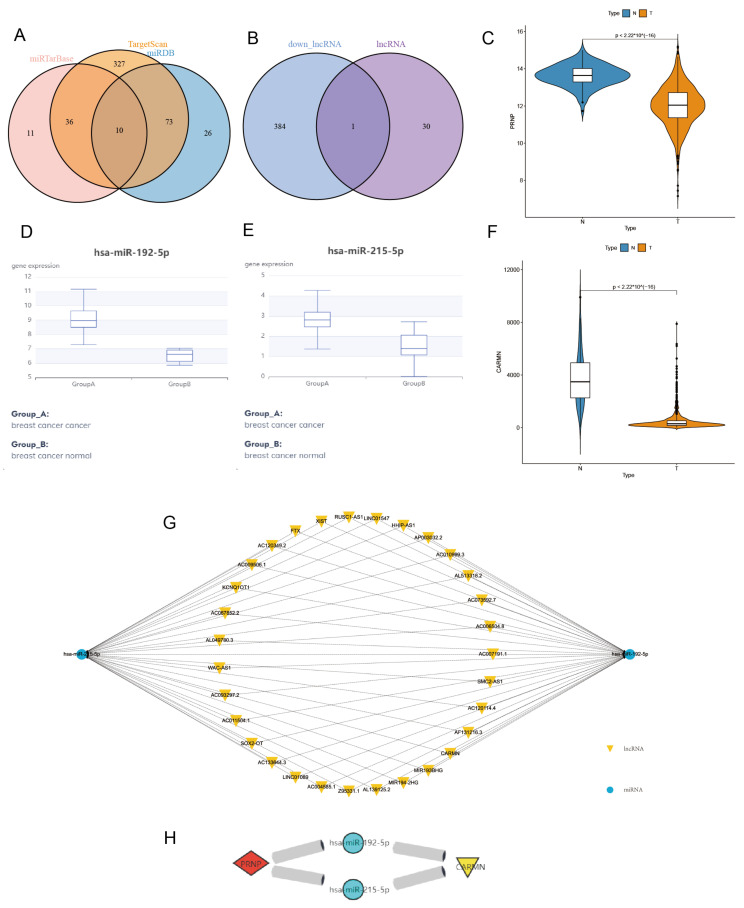
Construction and validation of ceRNA regulatory network in BRCA. (**A**) 10 PRNP-related miRNAs were identified, the intersection of TargetScan, miRDB, and miRTarBase database screening results. (**B**) The intersection of lncRNA (CRLs related with hsa-miR-192-5p and hsa-miR-215-5p) and down_lnRNA (down-regulated lncRNAs in TCGA-BRCA tumor samples), CARMN, a CRL that fit the ceRNA regulatory network hypothesis was identified. (**C**) Verify the expression of PRNP in TCGA-BRCA samples. (**D**) Verify the expression of the hsa-miR-192-5p and (**E**) hsa-miR-215-5p in breast cancer by the online database dbDEMC. (**F**) Verify the expression of CARMN in TCGA-BRCA samples. (**G**) miRNA-lncRNA regulatory network. (**H**) mRNA PRNP/miRNA hsa-miR-192-5p and hsa-miR-215-5p/lncRNA CARMN ceRNA regulatory network.

**Figure 7 diagnostics-13-01203-f007:**
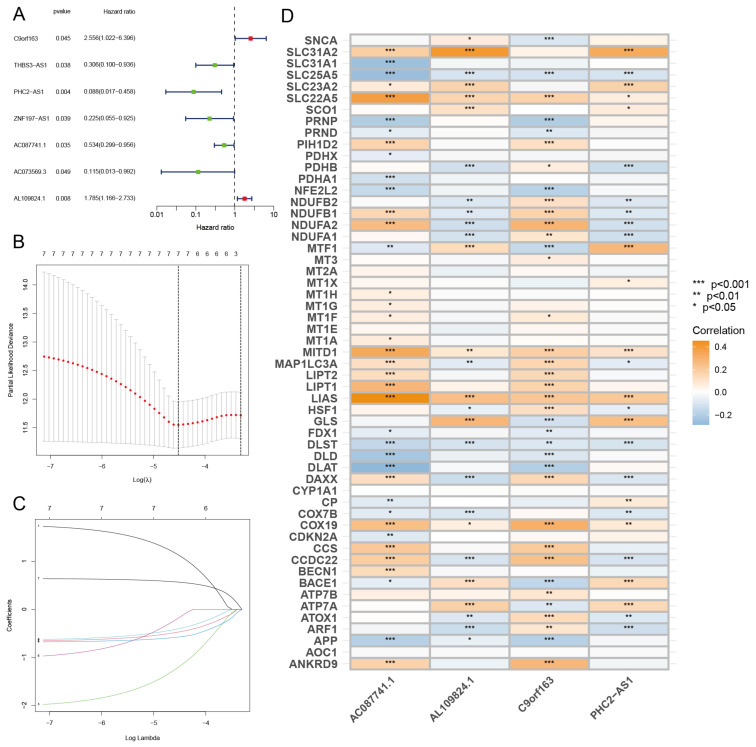
Univariate Cox, LASSO, and multivariate Cox regression analysis. (**A**) Forest plots of 7 prognostic CRLs. Red represents the HR value of risk CRLs and green represents the HR value of favorable CRLs. (**B**) Partial likelihood deviance of the 7 prognostic CRGs. (**C**) LASSO coefficients of the six prognostic CRLs. (**D**) Correlation heatmap of 57 CRGs and 4 CRLs. the darker the color, the more significant the correlation.

**Figure 8 diagnostics-13-01203-f008:**
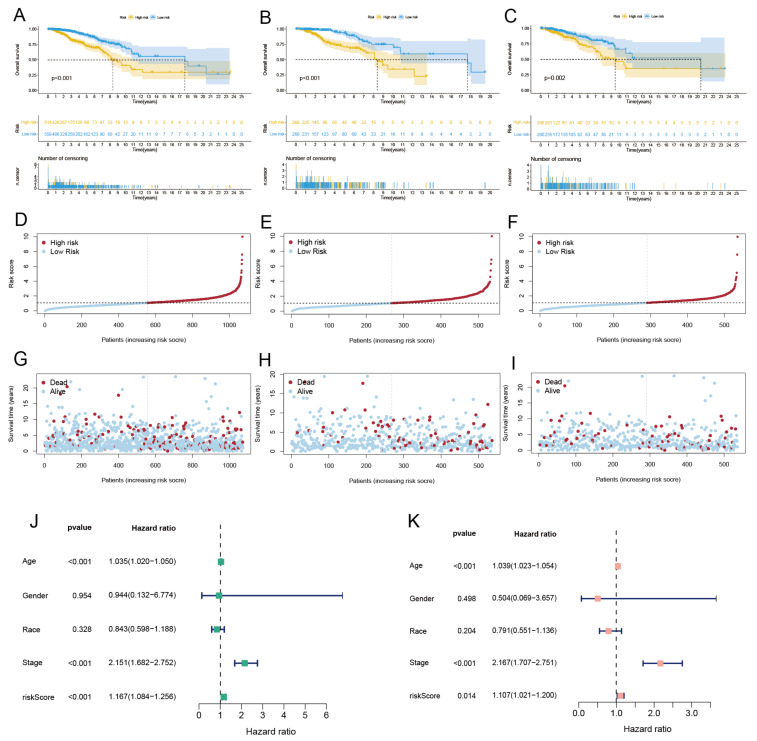
Survival curves and identified independent prognostic factors for high- and low-risk groups. (**A**) OS curves of all, (**B**) training, and (**C**) test samples in high and low groups of breast cancer patients. (**D**) Risk score distribution for all samples, (**E**) training samples, and (**F**) test samples of breast cancer. (**G**) Survival status of all samples, (**H**) training samples, and (**I**) test samples of BRCA in the high- and low-risk group. Different colors represent different states of existence. (**J**) Forest plots of risk models and clinical traits using univariate Cox and (**K**) multivariate Cox regression analysis.

**Figure 9 diagnostics-13-01203-f009:**
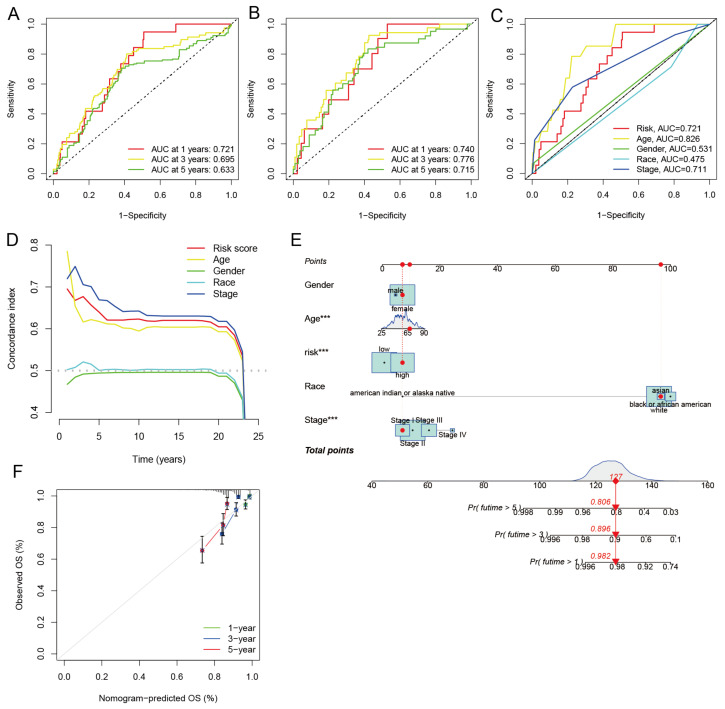
Assessment of the performance of the risk model and construction of the nomogram. (**A**) The ROC of 1-, 3-, and 5-year OS for all samples and (**B**) training samples. (**C**) The ROC for risk model and clinical traits. (**D**) C-index for risk models and clinical traits. (**E**) Construction of the nomogram by risk model and clinical trait. The red circle represents the patient’s score for different clinical traits, the red diamond represents the overall score, and the red triangle represents the patient’s 1-, 3-, and 5-year survival rate. (**F**) Calibration curves. ***, *p* < 0.001. *p* < 0.05 was considered statistically significant.

**Figure 10 diagnostics-13-01203-f010:**
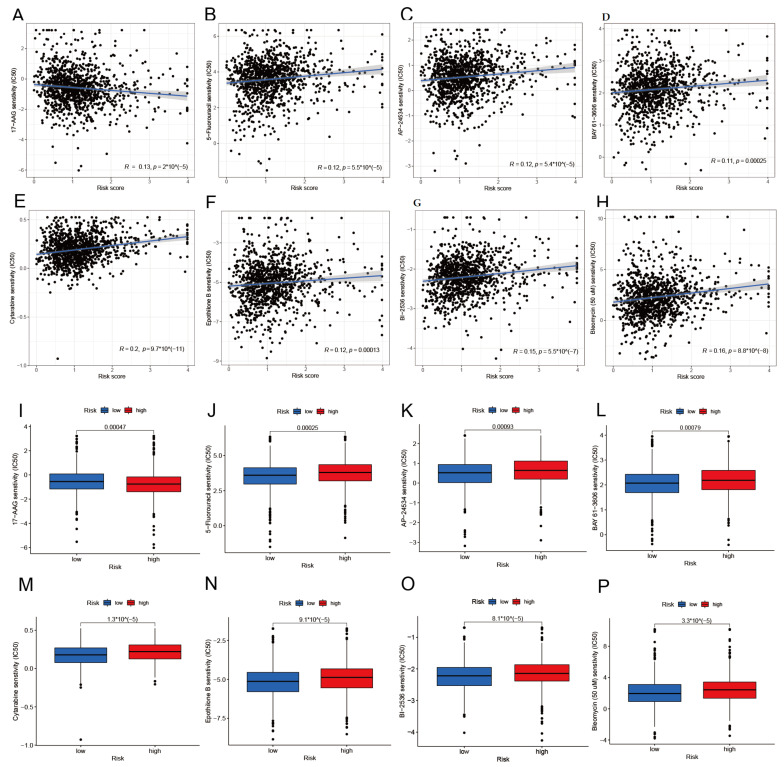
Screening for potentially sensitive medicines associated with BRCA. (**A**–**H**) 17-AAG showed a negative correlation with the risk score, while 5-Fluorouracil, AP-24534, BAY 61-3606, Cytarabine, Epothilone B, Bleomycin (50 uM), and BI-2536 showed a positive correlation with the risk score. (**I**–**P**) 17-AAG is significantly underexpressed in the high-risk group. Significantly overexpression of 5-Fluorouracil, AP-24534, BAY 61-3606, Cytarabine, Epothilone B, Bleomycin (50 uM), and BI-2536 was seen in the high-risk group.

## Data Availability

Data on corresponding BRCA normal and tumor samples were retrieved from GEO (https://www.ncbi.nlm.nih.gov/geo/, accessed on 16 September 2022) and TCGA (https://portal.gdc.cancer.gov/, accessed on 29 September 2022).

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
