# Peer review of "Construct ceRNA Network and Risk Model of Breast Cancer Using Machine Learning Methods under the Mechanism of Cuproptosis"

_diagnostics, 2023, doi:10.3390/diagnostics13061203_

Round 1

Reviewer 1 Report

The authors have implemented their proposed framework over the ceRNA network and risk model of breast cancer and the ML methods were used under the mechanism of cuproptosis.

Though the technicality of the claims presented are good, however the authors should consider the following suggestions to improve the overall quality of the manuscript:

1(1) The illustration of the identification of CRLs and construction of the risk framework should be further emphasized.

2(2) The Results section is well explained.

3(3) The work provided in depth discussion of the Deep Q Network and its mathematical derivation in terms of state-action function and their expected values.

4(4) The novelty of this work can be re-explained.

5(5) Further, the authors should consider improving the quality of Figure 1.

Overall presentation of the manuscript is good and should be accepted after incorporating the appropriate changes.

Reviewer 2 Report

A novel mechanism of programmed cell death called cuproptosis is linked to the development and spread of tumor cells. The function of cuproptosis in breast cancer is yet unknow to date. There are not machine learning studies to screen for characteristic genes to explore the role of cuproptosis-related genes (CRGs) in breast cancer. The authors proposed use 14 cuproptosis-related characteristic CRCGs that were discovered by the feature selection using the LASSO, SVM-RFE, and RF machine learning methods. The miRTarBase, TargetScan, and miRDB databases were used to identify hsa-miR-192-5p and hsa-miR-215-5p as the upstream miRNA of PRNP, and the upstream lncRNA, CARMN, was identified by StarBase database.

Through a series of bioinformatics analyses, the authors identified PRNP as a key CRCG, additionally they investigated the function and pathways of CRCGs enrichment and discovered four prognostic CRLs and 50 drugs related to breast cancer treatment.

The authors have performed a risk model with prognostic significance based on cuproptosis-related lncRNAs (CRLs). Also, novel ceRNA regulatory network and a novel risk model under the mechanism of cuproptosis have been designed that are the principal contribution of this study.

Comments:

1)     Majority of the Abbreviation presented in page 19 of this manuscript, but this reviewer thinks that it would be better also providing the abbreviations when you first use these ones. This reviewer did not find the abbreviations PRNP, ceRNA, SVM-RFE, ROC, etc when you used them first time.

2)     There are several small grammar errors, mainly commas and points. Please revise lines: 80, 138, 170, 192, 201-202, 388, 469.

Round 2

Reviewer 1 Report

Authors addressed the comments positively